# Severe cardiac and skeletal manifestations in *DMD*-edited microminipigs: an advanced surrogate for Duchenne muscular dystrophy
Masayoshi Otake [1,6] ✉, Michihiro Imamura [2,6], Satoko Enya[1], Akihisa Kangawa [1], Masatoshi Shibata[1], Kinuyo Ozaki[3,4], Koichi Kimura[5], Etsuro Ono[3,4] & Yoshitsugu Aoki [2] ✉

Duchenne muscular dystrophy (DMD) is an intractable X-linked muscular dystrophy caused by mutations in the *DMD* gene. While many animal models have been used to study the disease, translating findings to humans has been challenging. Microminipigs, with their pronounced physiological similarity to humans and notably compact size amongst pig models, could offer a more representative model for human diseases. Here, we accomplished precise *DMD* modification in microminipigs by co-injecting embryos with Cas9 protein and a single-guide RNA targeting exon 23 of *DMD*. The *DMD*-edited microminipigs exhibited pronounced clinical phenotypes, including perturbed locomotion and body-wide skeletal muscle weakness and atrophy, alongside augmented serum creatine kinase levels. Muscle weakness was observed as of one month of age, respiratory and cardiac dysfunctions emerged by the sixth month, and the maximum lifespan was 29.9 months. Histopathological evaluations confirmed dystrophin deficiency and pronounced dystrophic pathology in the skeletal and myocardial tissues, demonstrating that these animals are an unprecedented model for studying human DMD. The model stands as a distinct and crucial tool in biomedical research, offering deep understanding of disease progression and enhancing therapeutic assessments, with potential to influence forthcoming treatment approaches.

Duchenne muscular dystrophy (DMD) is an X-linked genetic disorder resulting from mutations in the *DMD* gene, leading to the absence of dystrophin in the muscle membrane. DMD is estimated to affect one in every 3500–5000 male births[1–3]. The disease is characterised by progressive muscle atrophy, which leads to cardiac and respiratory failure and, ultimately, premature death[4,5]. The clinical course of human DMD is severe and progressive, with muscle weakness typically appearing around the age of five years, loss of ambulation around 12 years, and death due to respiratory and cardiac failure in the late 30 s[6]. Several therapies aiming to partially restore dystrophin or tackle secondary pathologies have

gained regulatory approval, and numerous others are under clinical investigation[6]. But, there is an urgent need for more efficacious and definitive therapeutic interventions to provide curative outcomes for affected patients.

Animal models such as mice, rats, and dogs have been commonly used in preclinical research on DMD pathogenesis and drug development. While these models are practical, it is crucial to consider species-specific differences between humans and animals. Hence, a suitable surrogate that more accurately reflects human DMD is essential. Pigs have evolved as a valuable model due to their profound aptitude to replicate human diseases, allowing

[1]Swine and Poultry Research Center, Shizuoka Prefectural Research Institute of Animal Industry, Kikugawa, Shizuoka 439-0037, Japan. [2]Department of Molecular Therapy, National Institute of Neuroscience, National Centre of Neurology and Psychiatry, Tokyo 187-8502, Japan. [3]Department of Biomedicine, Graduate School of Medical Sciences, Kyushu University, Fukuoka 812-8582, Japan. [4]Center of Biomedical Research, Research Center for Human Disease Modeling, Graduate School of Medical Sciences, Kyushu University, Fukuoka 812-8582, Japan. [5]Departments of Laboratory Medicine/Cardiology, The Institute of Medical Science, The University of Tokyo, Tokyo, Japan. [6]These authors contributed equally: Masayoshi Otake, Michihiro Imamura. ✉e-mail: micropig@sp-exp.pref.shizuoka.jp; tsugu56@ncnp.go.jp

**Fig. 1 | CRISPR/Cas9-mediated generation of dystrophin-deficient microminipigs. a** CRISPR/Cas9-mediated gene-targeting strategy to generate dystrophin-deficient microminipigs. The target sequence and protospacer adjacent motif in the CRISPR/Cas9 genome editing system are indicated with a blue arrow and as PAM. The arrows in the diagram of the pig *DMD* gene indicate the positions of the promoters of the dystrophin isoforms. **b** Out-of-frame deletion of exon 23 of the microminipig *DMD* gene produced using the CRISPR/Cas9 gene-editing system. The numbers indicate the positions of the bases from the 5′ end of exon 23, which consists of 213 bases. **c** Three-month-old *DMD*-edited male microminipigs, F2-03 and F2-04. **d**, **e** A *DMD*-edited microminipig (F2-03) at 29 months of age. Oblique frontal standing view and supine view of anesthetised F2-03. The scale bar indicates 10 cm.

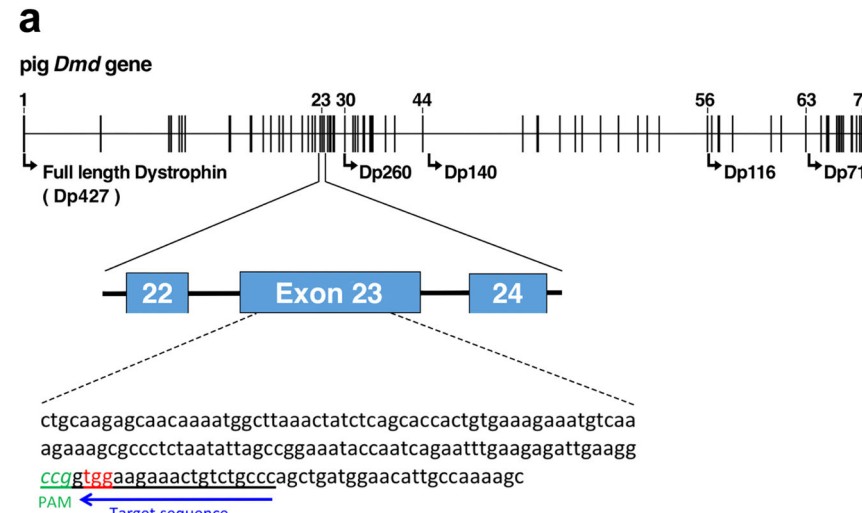

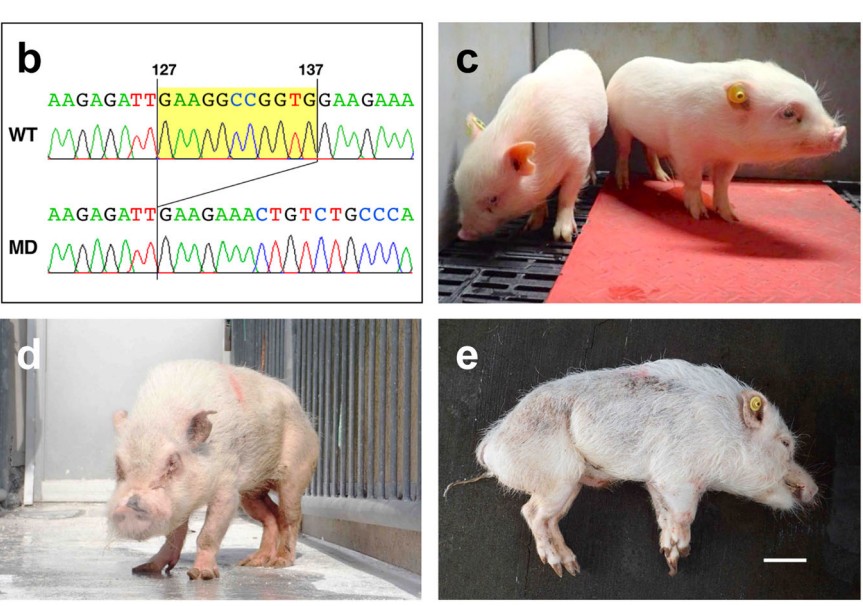

scientists to elucidate disease mechanisms and trajectories and appraise potential therapeutic measures attributable to their anatomical and physiological congruence with humans. Particularly, miniature pigs (minipigs) are gaining popularity because of their small size and ease of use in experiments. Regulatory agencies now recognise minipigs as valid non-rodent species for safety and efficacy studies as the scientific appropriateness of their use has been demonstrated.

The first DMD pig model was created by Klymiuk and co-workers, who used domestic pigs that displayed symptoms akin to human DMD. However, some pigs died shortly after birth, and those that survived were very short-lived and died within three months[7]. Subsequently, a DMD model was established in minipigs, which was found to be helpful in therapeutic studies but also showed a high mortality rate shortly after birth[8,9]. To address these limitations[7–9], a novel microminipig-based DMD model was developed. These microminipigs, derived from Vietnamese pot-bellied pigs at Fuji Micra, Inc. (Shizuoka, Japan) in 2010, are the world's most miniature laboratory pigs, weighing less than 25 kg at two years of age. Their small size allows easy handling, dosing, and cost-effective drug testing. Additionally, swine leucocyte antigens are well-defined, making them ideal for immunological reaction testing.

To enhance our comprehension of DMD and expedite its drug development, this study endeavoured to engineer microminipigs through the co-injection of embryos with the Cas9 protein and a single-guide RNA directed at exon 23 of *DMD*. This makes them highly suitable as models for studying human DMD and helps in predicting the potential effects of treatments and interventions in the disease with greater accuracy.

## Results

### Validation of a Cas9-single-guide (sg)RNA system targeting exon 23 of *DMD* in Cos-1 cells

The *DMD* gene in microminipigs contains 79 exons and encodes the muscle-specific dystrophin isoform. We designed a sgRNA specifically targeting exon 23 of *DMD* in microminipigs (Fig. 1a) based on a known mutation in exon 23 in *mdx* mice, which are a commonly employed model in DMD research[1]. The efficiency of the sgRNA was assessed by confirming the cleavage activity of the plasmid construct pX330/*DMD*_EX23, using the pCAG-EGxxFP system which contains the insert split EGFP[10]. The plasmids pX330/*DMD*_EX23 and pCAG-EGxxFP-*DMD*_Ex23 were co-transfected into Cos-1 cells, and enhanced green fluorescent protein (EGFP) fluorescence in the Cos-1 cells was visualised using fluorescence

microscopy. Minimal EGFP signals were observed in cells transfected with pCAG-EGxxFP-*DMD*_Ex23 alone or non-transfected cells, whereas clear signals were detected in co-transfected cells. These findings demonstrated that pX330/*DMD*_EX23 effectively cleaved exon 23 of *DMD*, validating the efficacy of the designed sgRNA in guiding the Cas9 protein to *DMD* exon 23 in Cos-1 cells.

## CRISPR-Cas9 zygote injection for generating *DMD*-edited microminipigs

To create out-of-frame mutations in *DMD* using the CRISPR/Cas9 system, we used the *DMD*_Ex23 sgRNA and Cas9 protein. Because CRISPR/Cas9-mediated gene editing in embryos can cause mosaic mutations, to ensure mutation stability and facilitate reliable phenotypic evaluation across lines, we tried to obtain F1 and F2 generations by crossing the resulting *DMD*-edited mutant pigs. A total of 30 embryos were surgically retrieved from three mated microminipig embryo donors. A mixture containing Cas9 protein (25 ng/μL) and *DMD*_Ex23 sgRNA (25 ng/μL) was injected into the cytoplasm of these embryos. The injected embryos were transferred into a surrogate female, resulting in the birth of three male offspring designated as F0-04, F0-05, and F0-06 after a gestation period of 114 days. Ear skin samples were collected from the three piglets to identify *DMD* modification.

Genomic DNA was extracted from the ear skins for PCR-based genotyping. The results revealed that the CRISPR/Cas9 system effectively disrupted the target site in F0-04, resulting in the deletion of 11 base pairs in exon 23 of *DMD* (Fig. 1b). However, no mutations were observed in F0-05 and F0-06. Mosaic mutations, a common occurrence in genetically modified mammals produced through CRISPR/Cas9 injection in zygotes, were responsible for the varied outcomes among the piglets. These mosaic mutations are attributed to sustained CRISPR/Cas9 effects after one-cell cleavage. Given these observations, it was imperative to further investigate mutations in other available samples, i.e. blood, of the three piglets. Genomic DNA was extracted and subjected to PCR-based sequencing. The results confirmed the presence of the same 11-bp deletion in exon 23, consistent with the findings in the ear skin samples, in F0-04.

## Off-target analysis of *DMD*-edited microminipigs

The potential for off-target effects is a persistent concern with the CRISPR/Cas9 technology. Therefore, we comprehensively analysed potential off-target mutations in the *DMD*-edited microminipigs. PCR products from 26 off-target sites in F0-04, F0-05, and F0-06 were subjected to DNA sequencing analysis. Remarkably, the results revealed no overlapping peaks or mutations at any of the off-target sites teste. These findings demonstrated that the CRISPR/Cas9 system employed in this study did not induce off-target genome mutations.

## Founder male selection and breeding of *DMD*-edited microminipigs

To identify a founder male among three *DMD*-edited microminipigs, genomic DNA was extracted from the sperm of F0-04, F0-05, and F0-06 for direct sequencing analysis. The 11-bp deletion was detected in the sperm of F0-04 but not in that of F0-05 or F0-06. Therefore, F0-04 was selected as the founder male. F0-04 was then mated with five wild-type (WT) females, resulting in the birth of 21 piglets (six males and 15 females). Eleven out of the 15 females harboured the 11-bp deletion, whereas this deletion was absent in all males. Therefore, further biological and clinical analyses were conducted using *DMD*-edited male pigs obtained by crossing these carriers with WT males (Fig. 1c–e).

## Characterisation of the dystrophin isoform and protein expression in the *DMD*-edited microminipigs

We initially examined the expression of dystrophin proteins in tissues of the *DMD*-edited microminipigs. Our genome editing approach resulted in an 11-bp deletion near the midpoint of *DMD* exon 23 (Fig. 1b). Reverse transcription (RT)-PCR analysis confirmed the deletion of the dystrophin transcript, which led to a frameshift mutation causing an alteration below

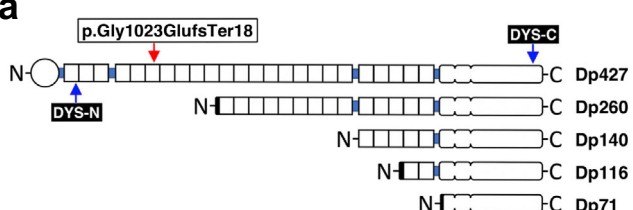

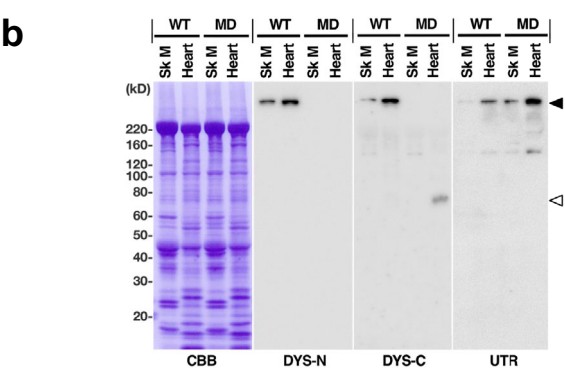

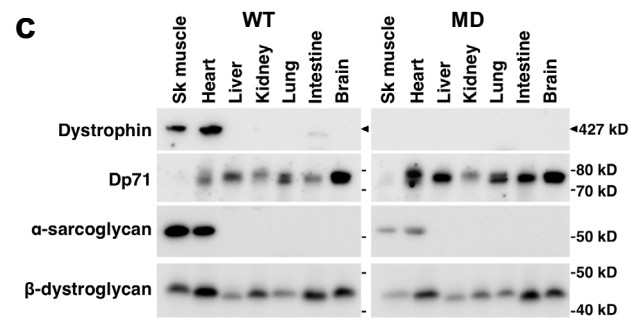

**Fig. 2 | Characterisation of dystrophin and dystrophin-associated proteins in *DMD*-edited microminipigs. a** Schematic representations of the dystrophin isoforms. N and C indicate the NH$_2$ and COOH termini of the proteins. Full-length dystrophin (Dp427) comprises an N-terminal actin-binding domain and a rod domain containing 24 spectrin-like repeats following the C-terminal domain associated with dystroglycan. The red arrow indicates the peptide termination site generated by genome editing. The blue arrows indicate the location of the epitopes recognised by the two antibodies used in this study (DYS-N and DYS-C). **b** Immunoblotting detection of dystrophin proteins and utrophin in the homogenates of skeletal muscle (Sk M) and hearts of WT and *DMD*-edited muscular dystrophy model (MD) pigs. The solid arrowhead indicates 427-kDa full-length dystrophin and utrophin (UTR), the open arrowhead indicates Dp71. Coomassie brilliant blue staining (left) revealed the proteins transferred on the membranes. Protein size markers are indicated on the left of the panel in kilodalton (kD). **c** Tissue expression of dystrophin and dystrophin-associated proteins in WT and MD microminipigs. Protein size markers are indicated on the right with dashes. An arrowhead indicates the molecular size of full-length dystrophin.

the glycine at position 1023, initiating a new reading frame starting at glutamic acid and terminating at position 18 (Figs. 1b, 2a). Immunoblotting using an antibody against the N-terminal region of dystrophin (DYS-N in Fig. 2a, b) revealed a full-length dystrophin band of 427 kDa in skeletal and heart muscle homogenates of WT pigs that was absent in corresponding homogenates of the *DMD*-edited pigs (Fig. 2b). Immunoblotting using an antibody against the C-terminus of dystrophin (DYS-C in Fig. 2a, b) also revealed the absence of full-length dystrophin in the edited pigs. A small band of approximately 70 kDa, representing dystrophin 71 (Dp71), was observed for heart homogenates from the *DMD*-edited microminipigs. This

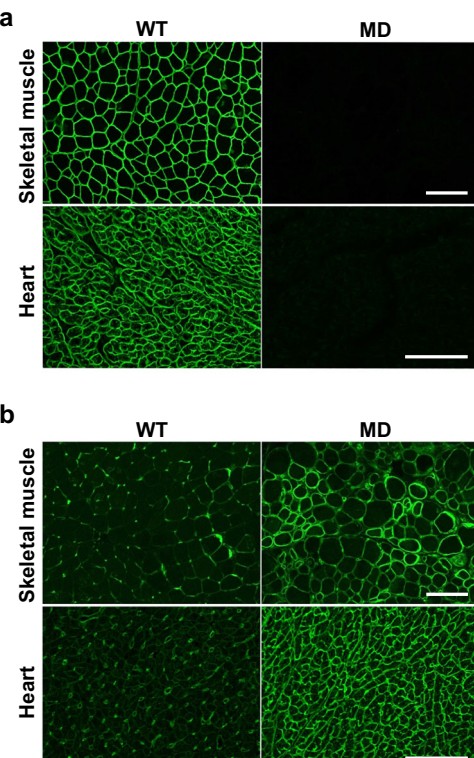

**Fig. 3 | Localisation of dystrophin and utrophin in skeletal and cardiac muscles of DMD-edited microminipigs.** Cryosections of the tibialis anterior muscle (Skeletal muscle) and inter-ventricular septum (Heart) were subjected to immuno-fluorescence staining with antibodies against the N-terminal region of dystrophin and the central region of utrophin. Panels (**a, b**) show full-length dystrophin and utrophin staining patterns, respectively. Bar, 100 μm.

antibody detected full-length dystrophin in the intestine and Dp71 in all tissues except skeletal muscle. Interestingly, the Dp71 signal was more robust in the *DMD*-edited microminipigs than in WT pig tissues, including the heart, liver, kidneys, and intestine (Fig. 2c). Further, we observed increased levels of utrophin, a dystrophin homologue, in the skeletal and heart muscles of the edited pigs (Fig. 2b). The levels of targeted pig β-dystroglycan and α-sarcoglycan, constituents of the dystrophin-associated protein complex, were reduced in the skeletal and heart muscles of edited pigs (Fig. 2c). These findings provide insights into the alterations in dystrophin and associated proteins in the *DMD*-edited microminipigs.

Next, we performed immunohistochemical analysis using specific antibodies to investigate the expression and localisation of the dystrophin isoforms and utrophin in *DMD*-edited microminipigs. Utilising the DYS-N antibody, we found that full-length dystrophin at the sarcolemma of skeletal and cardiac muscles was completely lost in the edited microminipigs (Fig. 3a). Conversely, the DYS-C antibody produced weak signals in the cardiac cell membranes of the *DMD*-edited pigs (Supplementary Fig. 1). The anti-utrophin antibody showed a noteworthy upregulation of utrophin, primarily at the sarcolemma, in the targeted pig muscles. In WT pig muscles, the same antibody predominantly stained capillary endothelial cells (Fig. 3b). These findings were in line with the immunoblotting results (Fig. 2b, c), providing consistent evidence of altered dystrophin and utrophin expression in the *DMD*-edited microminipigs.

## Clinical manifestations and mortality in *DMD*-edited microminipigs

*DMD*-edited microminipigs exhibited early-onset skeletal muscle weakness, displaying diminished and irregular locomotion, reduced momentum, and immobilisation beginning at one month of age. By three months of age, these animals showed difficulties in swallowing, drooling, thoracic

respiration, abnormal respiratory sounds, and occasional vomiting due to dysphagia evaluated by two veterinarians. Signs of diminished cardiac ejection fraction were evident at six months of age. Based on our findings that sperm formation in the testes and epididymis, and traces of ejaculation within the bred specimen's cage, we concluded that they reach sexual maturity at eight months of age. Their maximum lifespan was 29.9 months.

To characterise the *DMD* phenotype further, we assessed body weight differences between *DMD*-edited microminipigs (F2-03, F3-02, F3-03, F2-05, F2-13, F2-14) and WT controls (WT-01, WT-03, WT-04, WT-02, WT-05, WT-06, WT-10) (Supplementary Table 1). Until six months of age, there were no noticeable disparities in body weight (Fig. 4a); however, at around 10 months of age, the body weight difference between F2-03 and WT-01 became evident (Fig. 4a). Additionally, the serum creatine kinase level, a marker of muscle damage, was significantly elevated in the edited pigs (F2-10: 71,441 U/L at two months of age, F2-13 and F-14: 27,588 U/L and 29,507 U/L at six months of age, F3-02 and F3-03: 67,204 U/L and 34,868 U/L at 12 months of age, F2-03: 27,859 at 29 months of age) compared to that in the WT pigs (WT-09: 761 U/L at two months of age, WT-05 and WT-06: 1,425 U/L and 779 U/L at six months of age, WT-04: 5,962 at U/L at 11 months of age, WT-03: 591 U/L at 12 months of age, WT-01: 1,988 U/L at 29 months of age). The levels of other serum biochemical markers, such as aspartate aminotransferase and alanine aminotransferase, but not γ-glutamyl transpeptidase, were also elevated in the above *DMD*-edited pigs compared to those in WT pigs (Fig. 4b). Collectively, the pathological analyses indicated that *DMD*-edited microminipigs displayed characteristic phenotypes of DMD associated with dystrophin deficiency.

Alarmingly, six of the 13 *DMD*-edited microminipigs suddenly died between three and six months of age (F2-04 at three months, F2-05 at six months, F2-06 at four months, F2-07 at four months, F2-12 at three months, and F3-04 at three months of age). Our research holds significant implications for understanding the potential pathological mechanisms underlying the DMD-like manifestations observed in these animals[1,11]. For instance, F2-04 suddenly died around three months of age, possibly due to dyspnoea when forced to move.

## Histopathological characterisation of dystrophic muscle changes in *DMD*-edited microminipigs

We next comprehensively investigated the dystrophic muscle pathologies in *DMD*-edited microminipigs at different time points. At two months of age, we observed extensive skeletal muscle structural disruption, characterised by myofibre degeneration, necrosis, and regeneration in various skeletal muscles, including the diaphragm and tibialis anterior (Fig. 5a, Supplementary Fig. 2). However, pathological alterations were not evident in cardiac muscles at this age (Fig. 5b). By six months of age, the myocardium displayed slight degenerating cardiomyocytes and increased width of the interstitial connective tissue (Fig. 6a), which worsened by 12 months of age (Fig. 6b) according to haematoxylin and eosin (H&E) staining. Masson's trichrome staining revealed minimal fibrosis in the heart at six months (Fig. 6a), while substantial fibrosis was evident in the left and right ventricular and septum muscles at 12 months (Fig. 6b–d).

At 29.9 months of age, the pathological changes in the skeletal and cardiac muscles became more severe and apparent. The myofibre degeneration and regeneration observed at two months of age were no longer present. Instead, significant fibrosis was present in nearly all muscles, along with an increase in the width of intercellular spaces (Fig. 7a, Supplementary Fig. 3). Additionally, we noted marked myocyte swelling, self-destruction, and severe fibrosis in the left ventricular anterior and posterior wall (Fig. 7b). These observations improve our understanding of the progression of cardiac pathology in *DMD*-edited microminipigs, highlighting their potential as an informative animal model for studying DMD-related muscle pathologies. Pathological findings were not observed in the brain or other major organs. Our results shed light on the progressive dystrophic changes occurring in the skeletal and cardiac muscles of *DMD*-edited microminipigs, offering valuable information for future research in this field.

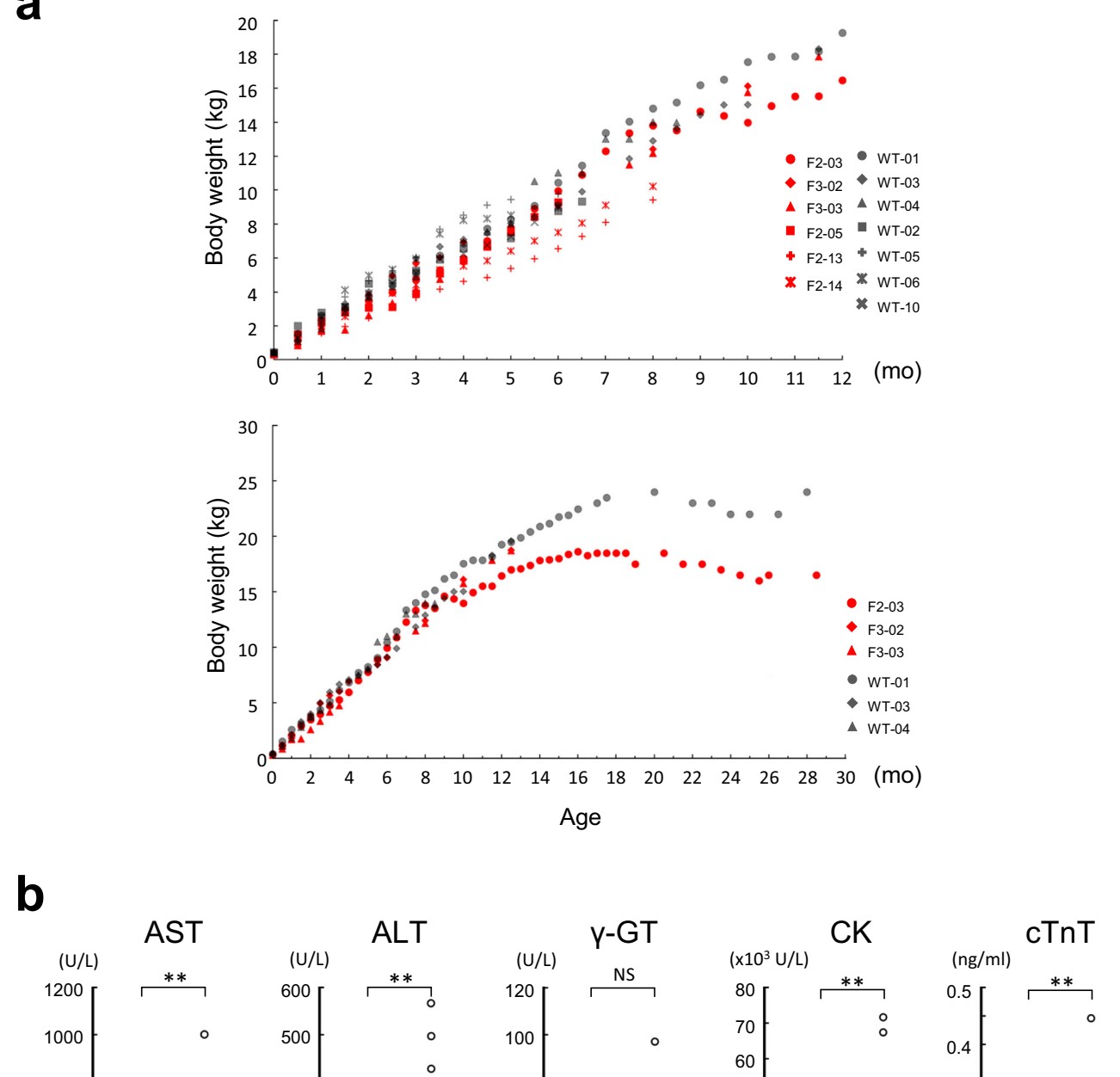

**Fig. 4 | Body weight and blood testing in *DMD*-edited microminipigs and WT controls. a** Body weight of *DMD*-edited microminipigs and age-matched WT pigs. Upper panel: body weights of six *DMD*-edited (F2-03, F3-02, F3-03, F2-05, F2-13, F2-14) and seven WT pigs (WT-01, WT-03, WT-04, WT-02, WT-05, WT-06, WT-10) from zero to 12 months of age. Bottom panel: body weights of three *DMD*-edited microminipig F2-03, F3-02, F3-03 and age-matched WT-01, WT-03, WT-04 from zero to 30 months of age. **b** Comparison of serum biochemical parameters between *DMD*-edited (MD) and WT microminipigs. MD; F2-03, F2-10, F3-02, F3-03, F2-13, F2-14. WT; WT-01, WT-03, WT-04, WT-05, WT-06, WT-09 (Supplementary Table 1). Aspartate aminotransferase; AST alanine aminotransferase, ALT γ-glutamyl transpeptidase, γ-GT creatine kinase, CK and cardiac muscle troponin T, cTnT. Significant differences (Student's *t* test) are indicated by asterisks. **$p < 0.01$.

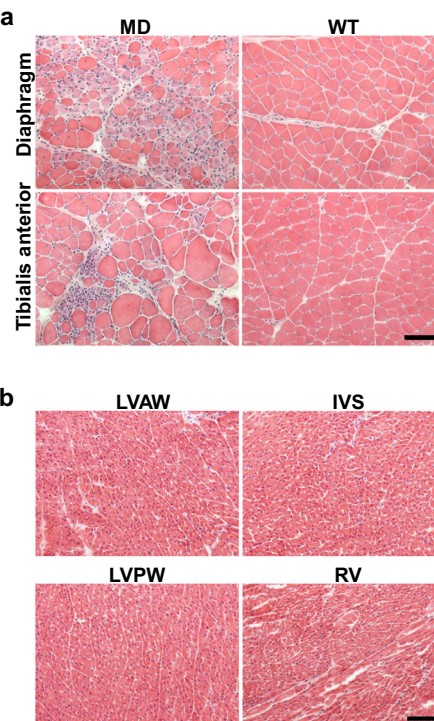

**Fig. 5 | Skeletal and cardiac muscle pathology in *DMD*-edited microminipigs at two months of age. a** H&E staining of diaphragm and tibialis anterior muscles of two-month-old DMD-edited microminipigs (MD) and WT microminipigs. **b** H&E staining of ventricular muscles of the microminipigs revealed no specific changes. Left-ventricular anterior wall, left-ventricular posterior wall, inter-ventricular septum, and right ventricle are indicated as LVAW, LVPW, IVS, and RV, respectively. Bar, 100 μm.

## Longitudinal assessment of the left ventricular ejection fraction in *DMD*-edited microminipigs

Finally, we compared cardiac phenotypes between WT and *DMD*-edited microminipigs by assessing the left ventricular ejection fraction (LVEF) using echocardiography at various time points (Table 1). At six months, the LVEF was 60.6% in WT (WT-10) and 52.3% in *DMD*-edited microminipigs, suggesting early cardiac impairment in the edited animals. At 11 or 12 months, WT-04 and WT-03 microminipigs exhibited LVEF values of 61.9% and 61.7%, respectively. These values are within the normal range. In contrast, two *DMD*-edited microminipigs (F3-03 and F3-02) displayed LVEF values of 45.9% and 52.0% respectively at 12 months, indicating a decline in cardiac function in the edited animals at this later age. These results demonstrated a progressive decrease in LVEF in *DMD*-edited microminipigs compared to that in WT pigs at six and 12 months. This is consistent with the clinical manifestations of cardiac impairment observed in human patients with DMD.

Our research outcomes offer significant insights into the pathophysiology of skeletal and cardiac muscular dystrophy in *DMD*-edited microminipigs, demonstrating their suitability as a viable model for translational and therapeutic research on DMD.

## Discussion

This report describes the successful generation of a DMD model using the CRISPR/Cas9 system in microminipigs. The *DMD*-edited microminipigs display characteristics consistent with human dystrophy, including the absence of the dystrophin protein, elevated serum creatine kinase activity, muscle fibre degeneration and regeneration, diminished locomotion, and sudden death resulting from respiratory failure. These findings indicate that the *DMD*-edited microminipigs hold great promise as an essential tool in advancing the development of therapeutic interventions for DMD.

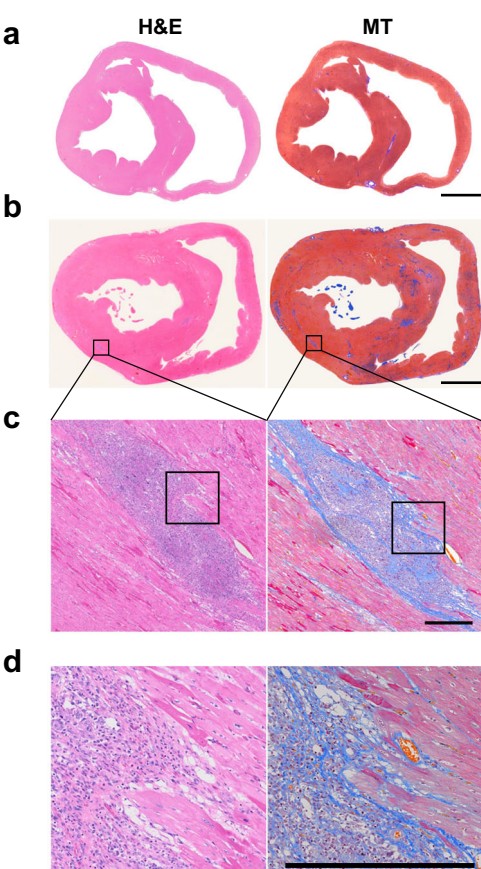

**Fig. 6 | Cardiac muscle pathology in *DMD*-edited microminipigs at six and 12 months.** H&E and Masson's trichrome (MT) staining of transverse sections of cardiac muscles of *DMD*-edited microminipigs at six months (**a**) and 12 months (**b**). Panels (**c, d**) are enlarged images of the rectangle areas in (**b, c**), respectively. Bars in (**a, b**), 1 cm, in (**c, d**), 500 μm.

Animal models have been instrumental in advancing our comprehension of muscular dystrophy pathogenesis and pave the way for therapeutic interventions[12–15]. However, it has to be acknowledged that these animal models exhibit species-specific clinical courses and phenotypic differences. The *DMD*-edited microminipigs generated in this study demonstrated a progressive decline in muscle strength, salivation, difficulties in swallowing, and vomiting. The F2-03 microminipigs displayed a crouched posture with hindquarter hyperflexion and forelimb abduction due to muscle atrophy, akin to phenotypic features of the canine X-linked muscular dystrophy (CXMD) model[16,17]. Additionally, F2-03 showed external ocular muscle degeneration, a characteristic not observed in the CXMD model[18]. These symptoms are comparable to those observed in DMD patients and CXMD dogs[16,17], indicating that *DMD*-edited microminipigs manifest pathological and functional features consistent with human disease.

The absence of dystrophin results in skeletal and myocardial muscle atrophies, potentially leading to cardiac or respiratory failures. Sudden death from dyspnoea was a prominent symptom in the *DMD*-edited microminipigs. However, no pathological changes were evident in the cardiac muscle at three months of age in F2-04. Considering the more severe destruction of myofibres in the diaphragm, it is likely that the sudden death of F2-04 resulted from diaphragm dysfunction, leading to dyspnoea. While a direct comparison between *DMD*-edited microminipigs and *mdx* mice is not feasible, similarities were noted in the overall phenotype, including degenerated and regenerated muscle fibres[19].

LVEF, a crucial indicator of cardiac function, progressively declines in DMD patients, ultimately leading to heart failure and premature mortality.

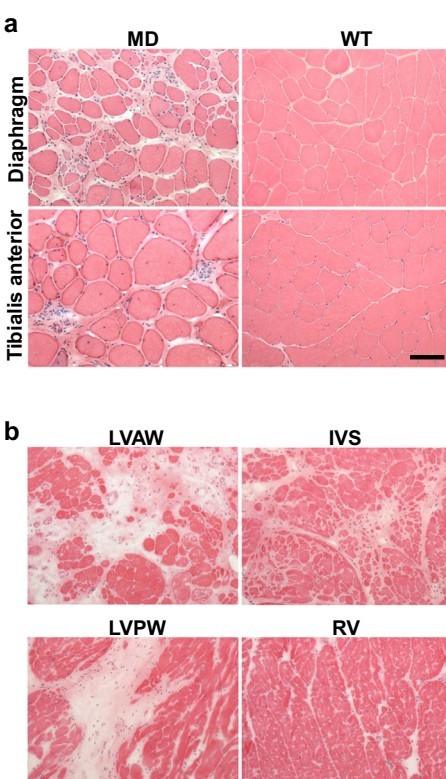

**Fig. 7 | Skeletal and cardiac muscle pathology in *DMD*-edited microminipigs at 29 months of age. a** H&E staining of diaphragm and tibialis anterior and (**b**) cardiac muscles at 29 months of age in *DMD*-edited microminipigs (MD) and WT microminipigs. Bar, 100 μm.

In healthy individuals, LVEF typically ranges from 60 to 70%, and a cut-off value of 55% is generally used to define normal LVEF in clinical settings. In the general population, LVEF values of 40% to 50% are categorised as mild, 30 to 40% as moderate, and below 30% as severe heart impairment. LVEF decreases over time in DMD patients, with values around 70% at around seven years of age, 60% at 12 years, 50% at 15 years, and 40% at 20 years. Around 40 years of age, LVEF declines to 20–30%, resulting in heart failure and an increased mortality risk. Our study revealed a progressive decline in LVEF in *DMD*-edited microminipigs compared to that in WT animals at six and 12 months, consistent with the clinical manifestations of cardiac impairment observed in human DMD patients. The age of 12 months in *DMD*-edited microminipigs corresponds to approximately 20 years in human DMD patients, and the LVEF is estimated to have declined to approximately 40% in humans of this age, aligning with the cardiac impairment observed in human patients with DMD.

Myocardial degeneration was not observed in F2-04 at three months of age, but particularly severe degeneration in the heart was evident in F2-03 at 29.9 months of age. This suggests that myocardial degeneration develops with ageing in *DMD*-edited microminipigs. In contrast, myocardial degeneration was seen after seven months in *mdx* mice harbouring a mutation in *DMD* exon 23[20], and myocardial wall fibrosis was not observed at 13 months of age in CXMD[21]. In a minipig model of DMD with modifications in *DMD* exon 52, most animals showed no histopathological abnormalities in the myocardium[7,8,22], but some exhibited myocardial abnormalities at nine months of age[23]. Cardiomyopathy is a significant cause of death in patients with DMD[24]. While further analysis is required to elucidate the mechanism underlying myocardial lesion progression in *DMD*-edited microminipigs, their suitability as a model for DMD is confirmed by the presence of muscular dystrophy and associated behavioural performance.

*DMD*-edited microminipigs were generated through crossbreeding with carrier female *DMD*-edited microminipigs. They exhibited behaviours indicative of sexual maturation identical to those of WT animals by eight months of age and had a maximum lifespan of 29.9 months. DMD minipigs have been successfully generated by somatic cell nuclear transfer (SCNT) using cells mutated by gene targeting or genome editing; however, most of them were short-lived. Yucatan minipig DMD models generated by gene targeting using recombinant adeno-associated virus died within the first week or were euthanised at seven months due to significant weight loss and severe symptoms beginning at two months of age[8]. Genome-edited Bama minipig DMD model animals all died within 12 weeks of birth[9]. The extended lifespan of the *DMD*-edited micromminipigs compared to previous minipig models highlights the potential utility of these microminipigs in investigating treatment options for DMD.

The different lifespan of the micromminipig from other pig models could be attributed to the generation of pig models by breeding. In fact, Wolf's group has successfully extended the lifespan of the pigs to the age of sexual maturity by changing the production of DMD model pigs from SCNT to crosses between heterozygous *DMD* mutation carrier females and wild-type males[23]. They also reported the importance of improved piglet management to reduce early mortality. The DMD pig production by SCNT showed varied weights of newborn pigs (about 400–2000 g), and there was a negative correlation between the birth weight and life expectancy[7]. In contrast, the birth weight of the DMD pig model produced by breeding was stable at around 1200 g, the boundary of early mortality[23]. The birth weight of the micromminipig model was about 400 g, which may be related to later survival. In the case of DMD dog model, genetic background was also suggested to affect their neonatal survival. Early in the process of creating a beagle-based DMD model colony by introducing semen from affected Golden Retrievers, the same carrier produced different types of dystrophic pups: neonatal fulminant or less severe dystrophic pups[25]. Furthermore, even in the same genetic background, it was reported that DMD mouse models (mdx) showed different peak timing of muscle regeneration activity depending on the mutation exons[26]. The peak of regeneration in exon 52 knockout mice (mdx52) was earlier than in mdx mice with the exon 23 mutation, indicating that myofiber degeneration begins earlier in mdx52. These observations suggested that relatively longer lifespan of *DMD*-edited micromminipigs was due to multiple factors, including body weight, improved neonatal management, genetic background, and the specific exon targeted.

In previous studies, it was observed that many piglets with *DMD*-deficiency died within a few days of their birth[7,8,23]. However, in our micromminipig model, only one piglet (mortality rate: 0.06%) died within five days of birth, aside from accidental deaths like crushing by the sow. This mortality rate is significantly lower than the previously reported rates in *DMD*-deficient piglets. An intriguing observation in the *DMD*-edited micromminipigs is their susceptibility to sudden death, known as porcine stress syndrome, occurring between three and six months of age. This phenomenon is also evident in other DMD pig models[23,27], suggesting that it is a characteristic symptom among DMD model pigs. Notably, *DMD*-edited micromminipigs had significantly lower body weights from one and a half to four months, when sudden death due to respiratory failure occurred, but no significant difference in growth from six to 12 months compared to WT animals.

Micromminipigs offer several advantages in biomedical research, as evidenced by their utilisation in various studies[28–31]. These advantages include their small size compared to laboratory miniature pigs and ease of handling, even after long-term rearing. The *DMD*-edited micromminipig emerges as an exceedingly advantageous animal model for several reasons. As adult animals weigh approximately 20 kg, they closely resemble the CXMD model[32], making them highly suitable for analysing the natural course of DMD, reducing administration costs, and enabling long-term observation of therapeutic effects.

While micromminipigs are a promising model for human DMD, the study has limitations and requires validation for broader DMD patient population. While our analysis did not reveal off-target mutations in the top 26 candidate off-target sequences, we acknowledge the inherent limitations of our screening methods. Consequently, we cannot categorically exclude

**Table 1 | Microminipigs used for echocardiography and the results obtained**

| Subject | Geno-type | Age (months) | Weight (kg) | HR (bpm) | IVS (mm) | LVEDd (mm) | PW (mm) | LVEF (%) | TMF-E (cm/s) | TMF-DT (ms) | TMF-A (cm/s) |
|---|---|---|---|---|---|---|---|---|---|---|---|
| WT-10 | WT | 6 | 9.1 | 124 | 5.9 | 24.2 | 4.6 | 60.6 | 60.9 | NA | 26.1 |
| F2-11 | DMD | 6 | 8.0 | 141 | 3.7 | 20.7 | 4.8 | 52.3 | 59.5 | 53 | 42.5 |
| WT-04 | WT | 11 | 16.4 | 101 | 5.6 | 27.8 | 5.7 | 61.9 | 25.6 | 45 | 44.7 |
| WT-03 | WT | 12 | 20.0 | 132 | 5.4 | 33.3 | 6.6 | 61.7 | 56.0 | 50 | 68.3 |
| F3-03 | DMD | 12 | 19.0 | 133 | 6.2 | 33.0 | 5.8 | 45.9 | 62.0 | 49 | 53.5 |
| F3-02 | DMD | 12 | 19.0 | 131 | 5.9 | 35.1 | 6.4 | 52.0 | 58.6 | 65 | 50.5 |

*DMD* 11-bp deletion in exon 23 of the *DMD* gene, *WT* wild-type, *A* peak atrial filling velocity, *DT* deceleration time of E wave, *E* peak early diastolic filling velocity, *HR* heart rate, *IVS* inter-ventricular septum, *LVEF* left ventricular ejection fraction, *LVEDd* left ventricular end-diastolic diameter, *PW* posterior wall, *TMF* transmitral flow velocity.

the presence of subtle off-target effects, such as inversions or mutations at non-predicted sites. Our findings suggest an absence of obvious off-target mutations within the analyzed sequences, though we recognize the potential for undetected subtle genomic alterations. Therefore, the long-term effects and potential off-target impacts of genetic modification require further research and understanding. It is essential to conduct a comprehensive analysis of the molecular mechanisms that underlie the observed phenotypes for future research. Additionally, researchers should aim to test potential therapeutic interventions in this model and evaluate their safety and efficacy profiles. This will establish a strong basis for future clinical translation and therapeutic advancements in human DMD.

In conclusion, we successfully established a *DMD*-edited microminipig model by genetically modifying pigs, targeting the *DMD* gene, using the CRISPR/Cas9 system. This microminipig model faithfully recapitulates phenotypic features of human DMD, making it a reliable and valuable tool for advancing research in DMD and accelerating the development of potential therapies.

## Materials and methods
### Animals
Thirty-four microminipigs and three regular-sized pigs used in this study were bred at the Swine and Poultry Research Centre, Shizuoka Prefectural Research Institute of Animal Industry (Shizuoka, Japan), and two microminipigs (F2-11 and WT-10) were bred at the General Animal Research Facility, National Institute of Neuroscience, National Centre of Neurology and Psychiatry (Tokyo, Japan) (Supplementary Tables 1 and 2). All animals were free from specific pathogens, including *Mycoplasma hyopneumoniae*, *Toxoplasma gondii*, *Erysipelothrix rhusiopathiae*, *Actinobacillus pleuropneumonia*, *Bordetella bronchiseptica*, *Pasteurella multocida*, *Haemophilus parasuis*, Suid herpesvirus 1, and porcine reproductive and respiratory syndrome viruses. The pig research protocols used were in accordance with the ARRIVE guidelines and the Guidelines for Proper Conduct of Animal Experiments (2006) issued by the Science Council of Japan, the representative organisation of the Japanese Scientist Community, and were approved by the Animal Care and Use Committees of Shizuoka Prefectural Research Institute of Animal Industry, Swine and Poultry Research Centre (H28-1, H29-1-3, R4-1-2 and R5-1), and National Institute of Neuroscience (2022030R1).

### Plasmid construction and transfection
CRISPR target sequences in exon 23 of the *DMD* gene were selected using CRISPRdirect (https://crispr.dbcls.jp/). pX330-U6-Chimeric_BB-CBh-hSpCas9 was purchased from Addgene[33]. *DMD*_Ex23_F (5′-caccGGG-CAGACAGTTTCTTCCAC-3′) and *DMD*_Ex23_R (5′-aaacGTGGAA-GAAACTGTCTGCCC-3′) oligonucleotide DNAs were annealed using the standard method. This short double-stranded DNA was inserted into the *Bbs*I restriction site in the pX330 vector using the DNA Ligation Kit (Takara Bio, Shiga, Japan), to generate pX330/*DMD*_EX23. For the EGxxFP system, part of exon 23 of *DMD* was amplified using TaKaRa Ex Taq™ (Takara Bio) and *DMD*_Ex23_F_NS (5′-CTGCAAGAGCAACAAAATGG-3′) and

*DMD*_Ex23_R_NS (5′-GCTTTTGGCAATGTTCCATC-3′) as the first primer pair, and *DMD*_Ex23_2nd_F (5′-GGCTTAAACTATCTCAG-CACCACTG-3′) and *DMD*_Ex23_R_NS as the second primer pair. The PCR product was purified using a FastGene® Gel/PCR Extraction Kit (Nippon Genetics, Tokyo, Japan) and inserted into pCAG-EGxxFP, which was obtained from Dr. Masato Ikawa, using Addgene plasmid # 50716 (http://n2t.net/addgene:50716; RRID: Addgene_50716) and a DNA Ligation Kit (Takara Bio). This plasmid was designated as pCAG-EGxxFP-*DMD*_Ex23. pX330/*DMD*_Ex23 and pCAG-EGxxFP-*DMD*_Ex23 were co-transfected into Cos-1 cells, isolated from the kidney of an African green monkey, using METAFECTENE PRO® reagent (Biontex, Munich, Germany). EGFP fluorescence in Cos-1 cells was observed using fluorescence microscopy (BX51, Olympus, Tokyo, Japan).

### Production of *DMD*-edited pigs by injecting zygotes with Cas9/sgRNA
Five mature female microminipigs (aged eight to 10 months) were used to obtain embryos, and three mature female Large White pigs were used as surrogate mother sows. All-female Superovulation was induced in the microminipigs by intra-muscular injection of 500 IU equine chorionic gonadotropin (eCG, PEAMEX®, SEROTOROPIN®; ASKA Animal Health, Tokyo, Japan) followed by 375 IU human chorionic gonadotropin (hCG, GESTRONE®1500; Kyoritsu Seiyaku, Tokyo, Japan) at a 72-h interval. Oestrus was induced in the female domestic pigs by intra-muscular injections of 1,000 IU eCG and 750 IU hCG[34]. Before eCG treatment, pseudo-pregnancy was generated in the eight- to 10-month-old mature female microminipigs and a surrogate mother sow by administering 0.1 mL/kg oestradiol benzoate (Ovahormondepot® intra-muscular injection 5 mg; ASKA Pharmaceutical) to the microminipigs and 4 mL for to a surrogate mother sow between nine and 12 days after oestrus, as previously described[35]. The day before eCG treatment, mature female microminipigs and a surrogate mother sow were intramuscularly injected with 1 and 2 mL of prostaglandin F2α, respectively, to synchronise the oestrus cycle[36]. The treated female microminipigs were mated with male microminipigs the day after hCG infection. Forty-eight hours after hCG injection, the female microminipigs were euthanised by exsanguination under deep anaesthesia, as described previously[37]. Single cell-stage embryos (19–23 h after mating) were surgically removed from the oviducts of the microminipigs by refluxing with 0.1% bovine serum albumin in phosphate-buffered saline and were immediately cultured in PZM-5 medium in the presence of 5% $CO_2$, 5% $O_2$, and 90% $N_2$ at 38.5 ℃[38]. An artificially synthesised sgRNA (GGGCAGACA-GUUUCUUCCACguuuuagagcuagaaauagcaaguuaaaauaaggcuaguccguuau-caacuugaaaaaguggcacggacucggugcuuuu) was purchased from FASMAC (Atsugi, Japan). Using a micro-injector (Narishige, Tokyo, Japan), 2–4 pL of a mixture of sgRNA (25 ng/μL) and Cas9 protein (25 ng/μL) (Guide-it™ Recombinant Cas9 Nuclease; Takara Bio) was injected into the cytoplasm of each embryo in PXM-HEPES medium under aerated conditions at 25–30 ℃[38]. The embryos were then transferred into the oviductal ampullae of an oestrus-synchronised surrogate mother sow. The surrogate mother sow was selected from among the pigs that showed the best oestrus sign following

hormone treatment, such as the immobility response to back pressure stimulation, and embryos were transferred into the tip of one side of the uterine horn using a glass Pasteur pipette. Anaesthesia was continued with 2–4% isoflurane (Isoflu®; Zoetis, Parsippany, USA) after using a combination of 0.1 mg/kg medetomidine (Domitor®; Nippon Zenyaku Kogyo, Tokyo, Japan) and 0.1 mg/kg midazolam (Dormicum Injection 10 mg; Maruishi Pharmaceutical, Osaka, Japan).

### Identification of mutations in the *DMD* gene and transcripts in microminipigs

Genome-edited mutations in the dystrophin gene and transcripts in microminipigs were identified by sequencing RT-PCR products around the sgRNA target site. Genomic DNA was extracted from skin, blood, or sperm collected from microminipigs using NucleoSpin® Tissue (Macherey-Nagel, Düren, Germany). Targeted fragments around the sgRNA target site were amplified as described above. Total RNA was prepared from the tibialis anterior inter-ventricular septum of the heart and cerebral cortex of *DMD*-edited microminipigs using TRI Reagent (Merck/Sigma-Aldrich, St. Louis, MO, US). RT-PCRs were run using a Takara RNA PCR Kit (Takara Bio, Shiga, Japan) and the following primers: *DMD*_Ex21_F (5′-GGCCAAA-GAGAAAGAGCTGC-3′) and *DMD*_Ex25_R (5′-AGACTGGGTT-GAATGGTCTGA-3′). PCR products were separated by 0.8% agarose gel electrophoresis and purified from the gel using a Monarch Gel Extraction Kit (New England BioLabs, Ipswich, MA, USA) or Wizard SV Gel and PCR Clean-Up System (Promega, Fitchburg, WI, USA). The purified PCR products were sequenced using the *DMD*_Ex23_2$^{nd}$_F primer for genomic DNA and *DMD*_Ex21_F and *DMD*_Ex25_R primers for mRNA (The Research Support Centre, Research Centre for Human Disease Modelling, Kyushu University Graduate School of Medical Sciences, Japan).

### Off-target assay

Potential off-target loci were searched using the CRISPR open tool (http://crispor.tefor.net/). The mismatch parameter for the off-target sequence was set to 5'-NGG-3' and chosen as PAM. Sites with less than three total mismatches were selected as potential off-target sites for testing. The predicted off-target sites are listed in Supplementary Table 3. The selected potential off-target sites were PCR-amplified using genomic DNA from the founder male as the template and the primers listed in Supplementary Table 4. The off-target sites were subjected to PCR-based sequencing as described above.

### Antibodies

Mouse monoclonal antibodies against N-terminal (34C5: Product code NCL-DYSB) and C-terminal (MANDRA1: Product number SAB4200763) dystrophin were purchased from Novocastra Laboratories (Newcastle upon Tyne, UK) and Merck/Sigma-Aldrich (Tokyo, Japan), respectively. Rabbit polyclonal antibodies against utrophin (UT-2), β-dystroglycan (β-DGs), and α-sarcoglycan (α-SG3) have been previously described[39]. Peroxidase-conjugated anti-mouse and anti-rabbit antibodies were purchased from Beckman Coulter (Brea, CA, USA) and Merck/Sigma-Aldrich. Alexa Fluor 488-conjugated anti-mouse and rabbit antibodies were purchased from Thermo Fisher Scientific KK (Tokyo, Japan).

### Immunoblotting

Sodium dodecyl sulphate (SDS) lysates of microminipig tissues were prepared from frozen tissue block sections. Appropriate amounts of cryosections (10 μm) were collected in centrifuge tubes and lysed in a Tris-buffered solution containing 2% SDS, 125 mM Tris-HCl (pH 6.8), 15% glycerol, and 5 mM dithiothreitol. After protein denaturation at 95℃ for 5 min, the solutions were centrifuged and the supernatants were collected as tissue lysates. Protein concentrations were determined with the Bio-Rad Protein Assay (Hercules, CA, USA), using bovine serum albumin as a standard.

Proteins were separated by SDS-polyacrylamide gel (4.5–15% gradient gel) electrophoresis and electrically transferred to Immobilon-P membranes (Merck Millipore, Burlington, MA, USA). After staining the proteins on the membranes with Coomassie Brilliant Blue dye solution, they were immunoreacted with primary antibodies (NCL-DYSB at 1:50, MANDRA1 at 1:600, UT-2 at 1:1000, β-DGs at 1:600, and α-SG3 at 1:500 dilution), followed by horseradish peroxidase-conjugated secondary antibodies. Immunoreactive protein bands were visualised using chemiluminescence detection reagent (Cytive, Marlborough, MA, USA) and recorded using a LAS-4000 image analyser (Fujifilm, Tokyo, Japan).

### Histological analyses

Tissues from three *DMD*-edited microminipigs (F2-04, F2-11, and F3-04) were collected post-mortem. Tissue samples from the other 12 microminipigs (F2-03, F2-10, F3-02, F3-03, WT-01, WT-03, WT-04, WT-07, WT-08, WT-09, and WT-10) were collected after euthanasia. All microminipigs except WT-10 received intra-muscular injections of 0.1 mg/kg Dormicum (5 mg/ml midazolam, Astellas Pharma, Tokyo, Japan) and 10 mg/kg ketamine (Fujita Pharmaceutical, Tokyo, Japan) as anaesthesia. WT-10 was given general anaesthesia by an inhalational mixture of 4% isoflurane and oxygen after intra-muscular injection of 0.5 mg/kg Dormicum and 0.1 mg/kg medetomine (medetomidine hydrochloride, Meiji Seika Pharma, Tokyo, Japan). After confirmation of the unconscious state of the animals by examining their state of immobility and insensibility to external stimuli: loss of both eyelid and corneal reflections and no response to skin pinching, they were euthanised by exsanguination[40,41]. Muscles and other tissues were immersed in Tragacanth Gum (Wako, Tokyo, Japan) and rapidly frozen in liquid nitrogen-cooled isopentane. Parts of these tissues were fixed in 10% formalin neutral buffer solution.

Cryosections were subjected to indirect immunofluorescent staining or H&E staining. Ten-micrometre-thick cryosections of the tibialis anterior muscle and inter-ventricular septum of microminipigs were placed on microscope slides and fixed in cold acetone. The fixed sections were equilibrated with Tris-buffered saline, blocked with Tris-buffered saline containing 2% casein, and reacted with primary antibodies (NCL-DYSB at 1:50, MANDRA1 at 1:600, and UT-2 at 1:1000 dilution) followed by Alexa 488-conjugated anti-mouse or anti-rabbit antibodies. Fluorescent signals in the muscle sections were observed with a confocal laser scanning microscope (Leica TCS SP5; Leica, Heidelberg, Germany). Cryosections (8 μm) of skeletal muscles (tibialis anterior muscle, intercostal muscle, extraocular muscle, longissimus dorsi, temporal muscle, soleus, rectus femoris, gastrocnemius, biceps brachii, and diaphragm) and ventricular muscles of microminipigs were placed on microscope slides, dried, and stained with H&E. All sections were photographed under a Leica DMR microscope.

Paraffin-embedded transverse sections of microminipig hearts were prepared and stained with Masson's trichrome and H&E staining at the Biopathology Institute (Oita, Japan). Stained sections were imaged using a SLIDEVIEW VS200 research slide scanner (Evident, Tokyo, Japan).

### Quantification of serum biomarkers

The levels of biomarkers, including aspartate aminotransferase, alanine aminotransferase, γ-glutamyl transpeptidase, creatine kinase, and cardiac muscle troponin T in serum samples from F2-03, F2-10, F3-02, F3-03, F2-13, F2-14 and WT-01, WT-03, WT-04, WT-05, WT-06, WT-09 were quantified using the Japan Society of Clinical Chemistry transferable method at Hoken Kagaku (Yokohama, Japan).

### Echocardiography

Microminipigs (F3-02, F3-03, WT-03, and WT-04) were anaesthetised using intra-muscular injections of 0.5 mg/kg Dormicum and 10 mg/kg ketamine (Fujita Pharmaceutical) and anaesthesia was maintained with an inhalational mixture of 2–2.5% isoflurane and oxygen. General anaesthesia in F2-11 and WT-10 was induced by intra-muscular injections of 0.5 mg/kg Dormicum and 0.1 mg/kg medetomine followed by maintenance with 2–2.5% isoflurane and oxygen. Echocardiography was performed in the left lateral decubitus position under above anaesthesia conditions using a Vivid E95 digital ultrasound system with a 12-MHz 12S-D sector probe (GE Healthcare Japan, Tokyo). In the parasternal long-axis view, inter-ventricular septum (IVS), posterior wall (PW), left-ventricular (LV) end-

diastolic diameter (LVEDd), and LV ejection fraction (LVEF) were calculated using the Teichholz method. In the apical four-chamber view, Doppler imaging variables of peak early diastolic filling velocity (E), deceleration time of E wave (DT), and peak atrial filling velocity (A) were measured.

## Statistics and reproducibility
Data are expressed as the mean ± standard deviation and were analysed using Student's $t$ test, considering between groups and significance at $p < 0.05$. *$p < 0.05$ and **$p < 0.01$ all represent values with significant difference using Pharmaco Basic Version 15.0.1 (Scientist Software, Tokyo).

## Reporting summary
Further information on research design is available in the Nature Portfolio Reporting Summary linked to this article.

## Data availability
The data supporting this study's findings are available from the corresponding author upon reasonable request. Images of uncropped blots are provided in Supplementary Fig. 4. Additional information associated with Fig. 4a, b are available in Supplementary Data 1.

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

## Acknowledgements
We thank M. Ikawa (Osaka University, Japan) for donating pCAG-EGxxFP plasmids. We are indebted to the staff at Shizuoka Prefectural Research Institute of Animal Industry, Swine and Poultry Research Centre for taking good care of the animals. This work was supported by the Japan Society for the Promotion of Science KAKENHI Grant-in-Aid for Challenging Exploratory Research (Grant Number 26660258) and an Intramural Research Grant (Grant number 2–6) from the National Centre of Neurology and Psychiatry.

## Author contributions
Masayoshi Otake: conceptualisation, methodology, investigation, data curation, writing – original draft, writing – review and editing, funding acquisition, project administration Michihiro Imamura: conceptualisation, methodology, investigation, data curation, writing – original draft, writing – review and editing Satoko Enya: investigation, data curation, visualisation, resources Akihisa Kanagawa: methodology, investigation, data curation, writing – original draft Masatoshi Shibata: supervision Kinuyo Ozaki: investigation, data curation, visualisation, resources Koichi Kimura: methodology, investigation, data curation Etsuro Ono: conceptualisation, methodology, investigation, data curation, writing – initial draft, writing – review and editing, supervision, funding acquisition, project administration Yoshitsugu Aoki: writing – review and editing, supervision.

## Competing interests
The authors declare no competing interests.
