## [Peer Review File · Communications Biology]

Reviewers' comments:

Reviewer #1 (Remarks to the Author):

This is a well-written manuscript on a new pig model for Duchenne muscular dystrophy. The novelty is at least two-fold: First, the genetic background is a micropig; second, a different exon has been targeted. The manuscript shows a first characterization of the model (blood chemistry, histology, heart function). It is interesting that these pigs, in part, achieve a longer lifespan than what is known from other porcine DMD models. Do you think this is solely due to reduced body weight, or could there be other influences at play? Was there increased mortality of DMD piglets around the time of birth, as described in other DMD pig models? Six DMD pigs spontaneously died around the 4th month of life (3-6 months). In the event of a therapeutic attempt, how would you choose the endpoint of the experiment to avoid premature loss of many involved animals? To maintain reasonable group sizes, considering the described fertility (21 piglets from 5 litters), a significant breeding herd would be necessary. What about inbreeding in the microminipig, given that, as mentioned in the initial publication, the breeding population traces back to only one animal (Catherin)? Do you think this, as described in the canine model for Duchenne muscular dystrophy (Kornegay, Joe N et al. "Golden retriever muscular dystrophy (GRMD): Developing and maintaining a colony and physiological functional measurements." *Methods in molecular biology* (Clifton, N.J.) vol. 709 (2011): 105-23. doi:10.1007/978-1-61737-982-6_7), could have an impact on the severity of the disease? You emphasize a similar body weight to the canine model, around 20 kg. Wouldn't a body weight closer to that of a human patient be preferable? Do you consider a combination of Midazolam and Ketamine suitable to ensure sufficient depth of anesthesia and pain suppression for performing euthanasia via exsanguination?

Reviewer #2 (Remarks to the Author):

Otake et al describe development and characterization of a new micromini-pig model for DMD. The pigs have a CRISPR-induced mutation in exon 23, similar to the original mdx mouse. Dystrophin is missing in muscles, although there is an interesting minor upregulation of Dp71. As in other models, utrophin is upregulated. This new model will likely prove to be enormously useful for DMD research and the development of novel therapeutics. The field has been clamoring for such a model for years. Mice are mildly affected, the canine models are often quite variable in their phenotype, and rats are not greatly larger than mice and have more cardiac than skeletal muscle issues. The previous pig models are either way too big to be used practically in a lab, or they have variable phenotypes and die young. While these new pigs also seem to suffer from some premature death, it does not appear as problematic as the existing mini-pig models. Hopefully these new pigs can be made available as I suspect they will be in great demand by other researchers and biotechs.

I have only two very minor comments. One, I think the authors were a little unclear in their comments about mosaic mutations on page 7. The phrasing implies that the non-edited pigs may be mutant in some tissues, in a mosaic pattern. I suspect the authors are simply saying that most knockouts will be somewhat mosaic in somatic tissues and must be outbred to the F1 generations to ensure Germaine heterozygosity. Second, on page 8 the authors declare that there are no off-target mutational events in the pig. While it seems true that none were found at the top 26 candidate off-target sequences, recent experiences suggest there could be other subtle off-target events such as inversions, or at random, non-predicted sites. I suggest softening the comment to say there appear to be no obvious off-target mutations.

Otherwise I think this is a wonderful manuscript and I congratulate the authors on their beautiful work

Reviewer #1:

Thank you for reviewing our manuscript on the new pig model for Duchenne muscular dystrophy (DMD). Your comments have been incredibly insightful, and we greatly appreciate the opportunity to address them. The changes we make based on your feedback will significantly improve the clarity and impact of our work.

- 1. It is interesting that these pigs, in part, achieve a longer lifespan than what is known from other porcine DMD models. Do you think this is solely due to reduced body weight, or could there be other influences at play?***

We agree that the longer lifespan of our micropig model, compared to other porcine DMD models, is interesting. While reduced body weight may contribute, other factors such as genetic background, the specific exon targeted, and improved husbandry practices may also play important roles. We have added sentences in the discussion section highlighted in red and new references as 26 and 27 (see pages 18-19, lines 293-311).

- 2. Was there increased mortality of DMD piglets around the time of birth, as described in other DMD pig models?***

Our data supports the fact that there is an increased mortality rate in DMD models, which is consistent with other observations in this field. The mortality rate of our DMD microminipig model was only 0.06%, which is considerably lower than previously reported rates in DMD-deficient piglets. We have highlighted this aspect more in the discussion section highlighted in red (see pages 19-20, lines 312-317).

- 3. Six DMD pigs spontaneously died around the 4th month of life (3-6 months). In the event of a therapeutic attempt, how would you choose the endpoint of the experiment to avoid premature loss of many involved animals?***

Thank you for your essential question. By studying the characteristics of this model animal and establishing breeding management based on our findings, we can significantly reduce the mortality rate in young animals aged between 3 to 6 months. We observed that individuals older than 6 months hardly experience any mortality. As a result, we suggest using animals older than 6 months for examining treatment methods to avoid unnecessary deaths. Moreover, we aim to explore the relationship between the development or causes of death and blood biochemical markers to analyze the connections with the causes of death in young animals. This will enhance our understanding and management strategies for these animals' welfare and survival.

4. *To maintain reasonable group sizes, considering the described fertility (21 piglets from 5 litters), a significant breeding herd would be necessary. What about inbreeding in the microminipig, given that, as mentioned in the initial publication, the breeding population traces back to only one animal (Catherin)?*

*Do you think this, as described in the canine model for Duchenne muscular dystrophy (Kornegay, Joe N et al. "Golden retriever muscular dystrophy (GRMD): Developing and maintaining a colony and physiological functional measurements." *Methods in molecular biology* (Clifton, N.J.) vol. 709 (2011): 105-23. doi:10.1007/978-1-61737-982-6_7), could have an impact on the severity of the disease?*

We understand that inbreeding can have a negative impact, which is a valid concern. We acknowledge that our model's limited genetic diversity stems from having only one ancestor, Catherin. To mitigate the effects of inbreeding, we have implemented strategies such as introducing new wild-type lines of micromini-pigs whenever possible.

5. *You emphasize a similar body weight to the canine model, around 20 kg. Wouldn't a body weight closer to that of a human patient be preferable?*

We have noted the relevance of comparing the body weight of canine models to that of human patients and agree that aligning more closely with human body weight could lead to further insights, particularly in therapeutic intervention studies. However, we would like to highlight the significant advantage of using our DMD microminipig models, which have a body weight of around 20 kg and are similar to the canine model. Compared to canine models, the comparable body weight of microminipigs offers researchers several benefits, such as lower therapeutic doses, reduced experimental costs, and less space required for housing. These benefits are briefly discussed in the conclusion section of our study.

6. *Do you consider a combination of Midazolam and Ketamine suitable to ensure sufficient depth of anesthesia and pain suppression for performing euthanasia via exsanguination?*

We appreciate your concern regarding the anesthesia and euthanasia methods. To ensure the best practices for animal welfare, we have chosen the combination of Midazolam and Ketamine, followed by exsanguination. This choice is based on established veterinary practices (Toxicol Pathol. 2019;47:469-482.) and ethical guidelines. We added a sentence describing how we confirmed the unconscious state of the animals under anesthesia with a reference (41) in the Materials and Methods section (see pages 29, lines 486-488).

We hope our responses and revisions to our manuscript address your concerns satisfactorily. We believe these changes have strengthened our study, and thank you again for your

constructive feedback.

Reviewer #2:

Thank you for your encouraging comments and constructive feedback on our manuscript. Our work is about developing and characterising a new micromini-pig model for Duchenne muscular dystrophy (DMD). Your insights confirm the potential impact of our work and help us refine our manuscript to ensure it is clear and accurate. We are pleased to address the two minor comments you raised.

- 1. I think the authors were a little unclear in their comments about mosaic mutations on page 7. The phrasing implies that the non-edited pigs may be mutant in some tissues, in a mosaic pattern. I suspect the authors are simply saying that most knockouts will be somewhat mosaic in somatic tissues and must be outbred to the F1 generations to ensure Germaine heterozygosity.*

We understand your concern about the clarity of our discussion on mosaic mutations. The intention was to convey that most CRISPR-induced knockouts could exhibit some level of mosaicism in somatic tissues, potentially complicating phenotype interpretation. Moreover, we aimed to highlight the necessity of breeding to F1 generations to achieve consistent germline transmission of the edited allele. To clarify this point, we have revised the section on page 7, lines 96-98, as follows:

"Because CRISPR/Cas9-mediated gene editing in embryos can cause mosaic mutations, to ensure mutation stability and facilitate reliable phenotypic evaluation across lines, we tried to obtain F1 and F2 generations by crossing the resulting DMD-edited mutant pigs."

- 2. While it seems true that none were found at the top 26 candidate off-target sequences, recent experiences suggest there could be other subtle off-target events such as inversions, or at random, non-predicted sites. I suggest softening the comment to say there appear to be no obvious off-target mutations.*

Thank you for your comment on our statement regarding off-target mutations. We agree that genome editing is complex and requires a more cautious interpretation. Although our comprehensive analysis did not identify any off-target mutations within the top 26 candidate sequences, we recognize the possibility of subtle off-target effects that may have been overlooked, such as inversions or mutations at unpredicted sites.

Therefore, we have updated our manuscript to reflect a more measured stance. Our statement now reads, "While our analysis did not reveal off-target mutations in the top 26 candidate off-

target sequences, we acknowledge the inherent limitations of our screening methods. Consequently, we cannot categorically exclude the presence of subtle off-target effects, such as inversions or mutations at non-predicted sites. Our findings suggest an absence of obvious off-target mutations within the analyzed sequences, though we recognize the potential for undetected subtle genomic alterations." (see page 21, lines 337-342)

Your suggestions have been invaluable in strengthening our manuscript, and we are grateful for your positive feedback. We hope our new DMD pig model will be a valuable resource for the research community and contribute significantly to advancing DMD research and therapeutics development.

REVIEWERS' COMMENTS:

Reviewer #1 (Remarks to the Author):

Thank you for addressing my comments.

A point-by-point response to the previous reviews

We are grateful for all the valuable feedback and suggestions. We have reviewed the document outlining the editorial requirements and will ensure that the final version of our submission meets all the necessary criteria. Below are our responses to the reviewers' comments (shown in italic font).

*OUR RESPONSES TO THE COMMENTS ARE SHOWN IN RED.

Issues

1. *For any Supplementary Files such as those mentioned above that are not included in your combined PDF (e.g. Supplementary Data, Movies, Audio, Software), please provide a title and description for each file here in the column to the right.*

"Supplementary Data" in Excel file, which includes Supplementary Figures 1-4.

"Supplementary Information" in PDF file, which includes Supplementary Tables 1-4 and additional information associated with Figures 4a and 4b.

2. *The abstract should be accessible to non-specialists and avoid jargon and abbreviations. Please write the abstract in the present tense.*

The abstract has been written in the present tense based on the direction.

3. *Editor's note: we suggest you remove table 1 and 2 from the main manuscript to improve readability. Please convert these tables to supplementary data files (excel file) and cite them within the manuscript accordingly.*

Tables 1 and 2 have been removed from the main manuscript. Those have been converted to supplementary data files (an Excel file) and cited within the manuscript accordingly.

4. *All blots/gels must be accompanied by size markers in every figure panel. Uncropped and unedited blot/gel images must be included as Supplementary Figure(s) in the Supplementary Information pdf.*

We confirm that all blots/gels are accompanied by size markers on every figure panel. Uncropped and unedited blot/gel images are included as Supplementary Figure(s) in the Supplementary Information pdf.